# Understanding the Effect of Triazole on Crosslinked PPO–SEBS-Based Anion Exchange Membranes for Water Electrolysis

**DOI:** 10.3390/polym15071736

**Published:** 2023-03-31

**Authors:** Jiyong Choi, Kyungwhan Min, Yong-Hwan Mo, Sang-Beom Han, Tae-Hyun Kim

**Affiliations:** 1Organic Material Synthesis Laboratory, Department of Chemistry, Incheon National University, Incheon 22012, Republic of Korea; 2Research Institute of Basic Sciences, Core Research Institute, Incheon National University, Incheon 22012, Republic of Korea; 3Boyaz Energy, 165 Gasandigital 2-ro, Geumcheon-gu, Seoul 08504, Republic of Korea

**Keywords:** anion exchange membrane, triazole, water electrolysis, phase separation

## Abstract

For anion exchange membrane water electrolysis (AEMWE), two types of anion exchange membranes (AEMs) containing crosslinked poly(phenylene oxide) (PPO) and poly(styrene ethylene butylene styrene) (SEBS) were prepared with and without triazole. The impact of triazole was carefully examined. In this work, the PPO was crosslinked with the non-aryl ether-type SEBS to take advantage of its enhanced chemical stability and phase separation under alkaline conditions. Compared to their triazole-free counterpart, the crosslinked membranes made with triazole had better hydroxide-ion conductivity because of the increased phase separation, which was confirmed by X-ray diffraction (XRD) and atomic force microscopy (AFM). Moreover, they displayed improved mechanical and alkaline stability. Under water electrolysis (WE) conditions, a triazole-containing crosslinked PPO–SEBS membrane electrode assembly (MEA) was created using IrO_2_ as the anode and a Pt/C catalyst as the cathode. This MEA displayed a current density of 0.7 A/cm^2^ at 1.8 V, which was higher than that of the MEA created with the triazole-free counterpart. Our study indicated that the crosslinked PPO–SEBS membrane containing triazoles had improved chemo-physical and electrical capabilities for WE because of the strong hydrogen bonding between triazole and water/OH^−^.

## 1. Introduction

The significance of sustainable energy sources is expanding, as is the demand for eco-friendly energy sources, as climate change is accelerating because of the indiscriminate use of fossil fuels. Hydrogen is one of the most efficient eco-friendly energy sources for storing significant amounts of energy since we can keep it in a high-density gaseous or liquid form. Besides being light, hydrogen has a higher energy density per kilogram (J/kg) than other fuels such as petroleum or graphite. These benefits allow hydrogen to store produced energy effectively [1,2,3].

Gray hydrogen, blue hydrogen, and green hydrogen are the three types of hydrogen sources that can be found [4,5,6]. Gray hydrogen is defined as hydrogen that has been extracted through the reaction of steam with methane, a key component of natural gas. This process now produces most of the hydrogen. Unfortunately, there is an excessive release of carbon dioxide while using this kind of hydrogen production process. The same process as gray hydrogen is used to produce blue hydrogen, but carbon dioxide capture and storage (CCS) technology is employed to limit the quantity of carbon dioxide produced. The highly developed CCS technology has drawn a lot of attention as a workable alternative, although it is still only partially effective in removing carbon dioxide from the atmosphere.

However, with the use of water electrolysis (WE), green hydrogen can be created without producing carbon dioxide. As a result, this kind of hydrogen holds promise as a future clean energy source. However, WE’s low hydrogen production efficiency necessitates high power consumption, which makes commercialization challenging [5,7]. Recent years have seen numerous studies on WE to produce green hydrogen as the most environmentally benign energy source [4,8,9].

Alkaline water electrolysis (AWE), proton exchange membrane water electrolysis (PEMWE), and anion exchange membrane water electrolysis (AEMWE) are the three types of WE procedures based on the electrolyte employed [10,11,12]. AWE is a technique that uses alkaline solutions as electrolytes to electrolyze water to produce hydrogen and oxygen. Although this method has received more research than other alternatives, it is known that using porous membranes in AWE might cause hydrogen and oxygen crossing, lowering the purity of the produced hydrogen [13].

Contrarily, PEMWE produces high-purity hydrogen by using a proton exchange membrane (PEM) as an electrolyte. However, the application of PEMWE is less advantageous because of the use of noble-metal catalysts, such as Pt, as well as the high costs of cell components, including perfluorosulfonic acid-based polymer electrolyte membranes, for example, Nafion [14,15,16].

PEMWEs have recently been replaced by promising anion exchange membrane-based (AEM)-based AEMWEs [8,17,18]. Because the oxygen evolution reaction (OER) occurs under alkaline conditions in AEMWEs, non-platinum metal catalysts, such as Ni- and Co-based catalysts, may be used in place of PEMWEs as oxygen evolution reaction (OER) catalysts because they are less expensive. Moreover, liquid electrolytes are not employed, in contrast to AWE. Instead, solid electrolytes, such as polymer electrolyte membranes, are used, making it possible to manufacture high-purity hydrogen. However, because OH^−^ ions are conducted instead of H^+^ ions in an AEMWE, its ionic conductivity is lower than that of PEMWEs. The performance of the cell may also be affected by this decreased ionic conductivity. Moreover, the AEM’s polymer backbone and ion-conducting groups are vulnerable to OH^−^ ion breakdown, which reduces durability. To get beyond these restrictions, high ionic conductivity and chemical stability AEMs must be developed [8,17,19].

A polymer backbone, which alters the characteristics of the membrane, and ion-conducting head groups, which conduct OH^−^ ions, comprise a polymer electrolyte appropriate for use as an AEM [20,21]. Both aryl ether-type and non-aryl ether-type polymers have been employed as polymer backbones, including poly(arylene ether ketone) (PEK) [22,23,24], poly(arylene ether sulfone) (PES) [25,26], poly(phenylene oxide) (PPO) [27,28,29,30], styrene ethylene butylene styrene (SEBS) [27,31,32,33,34,35], polyphenylene (PP) [12,32,33,36,37], and polyethylene (PE) [38,39,40,41]. Ion-conducting groups are often quaternary ammonium [12,32,33,34], morpholinium [42,43], pyridinium [32,33,44], imidazolium [45,46], phosphonium [47,48], and metal coordination compounds [49]. In particular, the most popular aryl ether-type and non-aryl ether-type polymer backbones are PPO and SEBS because they are commercial polymers with respectable physicochemical features. Ease of synthesis and high conductivity make quaternary ammonium one of the most popular ion-conducting head groups. Several AEM materials showing comparable or even higher cell performances than the PEMWE have recently been reported. These include polycabazole [50] and poly(fluorenyl-co-aryl piperidinium) (PFAP) [51].

A PPO-Tri-SEBS-based crosslinked polymer (**A**) as a novel AEM material has recently been produced by employing the triazole group to induce crosslinks in PPO and SEBS. The resulting polymer showed good AEM fuel cell (AEMFC) performance (Figure 1) [27]. Triazole was used to chemically link two polymers with contrasting mechanical characteristics, PPO and SEBS, creating a crosslinked polymer with exceptional tensile and elastic capabilities. Moreover, when reacted with water, the triazole group improved the ability of the membrane to retain water, leading to outstanding fuel cell performance. It appears that the effects of hydrogen bonding from a triazole group can improve the overall fuel cell performance of the AEM. Based on this concept, the effects of the same triazole phenomenon on WE, where the cell operates in water, not under humid conditions, were examined. That is, instead of focusing on room humidity, the current study concentrated on the impact of the interaction of the triazole with water during WE. To accomplish this, a PPO–SEBS membrane (**B**) was created by crosslinking PPO and SEBS without the addition of triazole, and its characteristics were compared to those of the PPO-Tri-SEBS membrane. To increase the water uptake (WU) of the manufactured membrane, the acylation degree of SEBS in the earlier study mentioned above was set to 70%. However, the membrane produced for the present study was designed for operation in water rather than humid conditions; hence, the degree of acylation was reduced to 50%. The membranes x-TriPPO-50SEBS and x-PPO-50SEBS were produced as a result (Figure 1). The WE performance of these two crosslinked membranes and their physicochemical and electrochemical characteristics were evaluated.

## 2. Materials and Methods

### 2.1. Materials

Asahi Kasei Co. (Tokyo, Japan) provided the poly (2,6-dimethyl-1,4-phenylene oxide) (PPO) (M_n_ = 34 000 g mol^−1^, M_w_ = 59,000 g mol^−1^). Kraton (Houston, TX, USA) was requested to provide Poly(styrene-b-ethylene-co-butylene-b-styrene) (SEBS, A1533H) with a 57% styrene content. Trimethylamine solution (~45 wt% in H_2_O), copper(I) bromide (CuBr(I), 98%), chlorobenzene (99.5%), 6-bromohexanoyl chloride (97%), aluminum chloride (AlCl_3_), and 3-dimethylamino-1-propyne (97%) were all purchased from Sigma–Aldrich (St. Louis, MO, USA). The following substances were bought from TCI (Tokyo, Japan): N-bromosuccinimide (NBS, 98%), sodium azide (NaN_3_, 99%), N,N,N′,N″,N″-pentamethyldiethylenetriamine (PMDTA, 99%), and dimethylamine (approx. 10% in THF). Alfa-Aesar (Haverhill, MA, USA) provided triethyl silane (98%) for purchase. Alternative commercial providers were used to obtain other chemicals that were not mentioned above. Throughout the investigation, deionized (DI) water was used for membrane treatment and property evaluations.

### 2.2. Synthesis of Bromobenzylated PPO (Br-PPO) 3

In a 100 mL two-neck round-bottom flask equipped with a reflux condenser and a magnetic stirrer and operating in a nitrogen atmosphere, PPO (5 g, 42 mmol) was dissolved in chlorobenzene (60 mL). N-bromosuccinimide (4.5 g, 25 mmol) and azobisisobutyronitrile (0.2 g, 11.7 mmol) were added to the mixture after the polymer had completely dissolved. The mixture was then rapidly agitated, slowly heated to 135 °C, and maintained at this temperature for 4 h. We placed the reaction mixture in a sizable volume of methanol (1000 mL) after being allowed to settle to ambient temperature, where the polymer precipitated. To remove any leftover reactants, the precipitated polymer was filtered and washed repeatedly in methanol. After the polymer was dried at 80 °C in a vacuum oven for 24 h, it produced the bromobenzylated PPO (Br-PPO, **3**), which was obtained as a brownish solid (6.2 g, 99.0%); H (400 MHz, CDCl_3_) 6.76–6.63 (1.84H, broad signal, H_4_), 6.58–6.44 (3.73H, broad signal, H_3_), 4.35 (2H, s, H_2_) (14.56H, broad signal, H_1_).

### 2.3. Synthesis of Azidobenzylated PPO (N_3_-PPO) 4

In a 100 mL two-neck round-bottom flask with a magnetic stirrer and a nitrogen atmosphere, Br-PPO 3 (5 g, 11 mmol) was dissolved in NMP (65 mL) to create a homogeneous solution. After the polymer had completely dissolved, sodium azide (3.7 g, 57 mmol) was added to the mixture. Over 24 h, the reaction mixture was gradually heated to 60 °C. The mixture was put into a sizable volume of methanol (1000 mL) and cooled to room temperature once the reaction was finished, which caused the polymer to precipitate. The precipitated polymer was filtered and repeatedly washed with methanol to remove any remaining reactants. To obtain the azidobenzylated PPO (N_3_-PPO, 4) as a brownish solid (3 g, 93.5%), the polymer was next dried in a vacuum oven for 24 h at 80 °C; δ_H_ (400 MHz, CDCl_3_) 6.75–6.60 (19.44H, broad signal, H_4_), 6.48 (62.39H, broad signal, H_3_), 4.22 (20.00H, s, H_2_), 2.10 (224.97H, broad signal, H_1_).

### 2.4. Synthesis of Dimethyl Amine-Triazole-Functionalized PPO (DMA-Tri-PPO) 1

In a 100 mL Schlenk flask, N3-PPO 4 (3 g, 7 mmol) was mixed in NMP (20 mL) to create a homogenous solution. N,N,N′,N″,N″,N″-pentamethyl diethylenetriamine (1.2 mL, 6 mmol), copper(I) bromide (400 mg, 2 mmol), and N,N-dimethylprop-2-yn-1-amine (1.2 mL, 11 mmol) were added after the polymer had completely dissolved. With the use of liquid nitrogen, three freeze-thaw cycles were used to rid the solution of oxygen. The reaction mixture was heated to 50 °C for 24 h while being vigorously agitated. After the reaction was finished, the liquid was cooled to room temperature and placed into a 1000 mL, 3:1 *v/v* mixture of DI water and methanol. The polymer precipitated out and was filtered and washed repeatedly with DI water and methanol to eliminate any residual reactants. The polymer was then dried for 24 h at 80 °C in a vacuum oven to produce the undesirable dimethyl amine-triazole-functionalized PPO (DMA-Tri-PPO, **1**), which was obtained as a brownish bead (2.6 g, 86.6%); *δ*_H_ (400 MHz, CDCl_3_) 7.81–7.48 (10.11H, broad signal, H_8_), 6.64 (9.47H, broad signal, H_7_), 6.48 (42.05H, broad signal, H_6_), 5.38 (20.00H, broad signal, H_5_), 3.63 (20.25H, broad signal, H_4_), 2.30 (61.42H, broad signal, H_3_), 2.10 (139.49H, broad signal, H_1–2_).

### 2.5. Synthesis of Dimethylamine-Functionalized PPO (DMA-PPO) 2

In a 100 mL two-neck round-bottom flask with a magnetic stirrer and a nitrogen atmosphere, Br-PPO **3** (2 g, 4 mmol) was dissolved in NMP (20 mL), forming a homogenous solution. Dimethylamine (6.5 mL, 13 mmol) was added to the mixture when the polymer had completely dissolved, and the reaction was gradually heated to 45 °C for 24 h. The mixture was put into a sizable volume of methanol (1000 mL) and cooled to room temperature once the reaction was finished, which caused the polymer to precipitate. To remove any leftover reactants, the precipitated polymer was filtered and washed repeatedly in methanol. The polymer was then dried for 24 h at 80 °C in a vacuum oven to create the dimethylamine-functionalized PPO (DMA-PPO, **2**), which was produced as a brownish solid (1.8 g, 95.6%); *δ*_H_ (400 MHz, CDCl_3_) 6.78 (1.00H, broad signal, H_5_), 6.59–6.39 (4.79H, broad signal, H_4_), 3.49–3.23 (2.40H, s, H_3_), 2.26 (6.00H, broad signal, H_2_), 2.10 (14.97H, broad signal, H_1_).

### 2.6. Synthesis of Bromohexanoyl SEBS(Br-Hex-CO-SEBS) 6

Dichloromethane (150 mL) was used to dissolve SEBS (5 g, 27 mmol) to create a homogeneous solution in a 500 mL round-bottom flask fitted with a magnetic stirrer in a nitrogen-filled environment. 6-Bromohexanoylchloride (2.8 mL, 18 mmol) and aluminum chloride (2.4 g, 18 mmol) were gradually added using a dropping funnel and stirred at room temperature once the polymer had completely dissolved. After 24 h, the reaction mixture was put into 1000 mL of methanol, which precipitated the polymer. To exclude any remaining reactants, the precipitated polymer was filtered and washed numerous times with methanol. The polymer was then dried for 24 h in a desiccator, yielding a white solid (7.1 g, 99.2%) with a 50% molar ratio of bromohexanoyl SEBS (Br-Hex-CO-SEBS, **6**); *δ*_H_ (400 MHz, CDCl_3_) 7.96–7.39 (2.25H, broad signal, H_17–18_), 7.26–6.26 (7.96H, broad signal, H_14–16_), 3.43 (2.00H, broad signal, H_13_), 3.05–2.78 (2.00H, broad signal, H_12_), 2.70–0.42 (40.17H, broad signal, H_1–11_).

### 2.7. Bromohexyl SEBS (Br-Hex-SEBS) 5 Is Created through the Reduction of Br-Hex-CO-SEBS 6

Br-Hex-CO-SEBS **6** (7.2 g, 12 mmol) was homogeneously dissolved in chloroform (150 mL) in a nitrogen-filled 500 mL round-bottom flask with two necks. Trifluoroacetic acid (TFA) (29 mL, 409 mmol) and triethyl silane (20 mL, 123 mmol) were added after the polymer had completely dissolved. The reaction mixture was continuously stirred for 48 h while being gradually heated to 105 °C. Following the cooling of the reaction mixture to room temperature, 100 mL of 1 M KOH was added to neutralize any remaining TFA. The organic layer was then put into 1000 mL of methanol, the precipitated polymer was filtered, and any leftover reactants were washed away with methanol by numerous rounds of methanol washing. Lastly, the produced polymer was vacuum-dried in a desiccator for 24 h. The target product, bromohexyl SEBS (Br-Hex-SEBS, **5**), was produced by drying the obtained polymer at room temperature in a vacuum for 24 h. It is a white solid (6.8 g, 98.4%); *δ*_H_ (400 MHz, CDCl_3_) 7.21–6.17 (9.27H, broad signal, H_15–19_), 3.40 (2.00H, broad signal, H_14_), and 2.65–0.55. (41.66H, broad signal, H_1–13_).

### 2.8. Fabrication of Crosslinked x-TriPPO-50SEBS A and x-PPO-50SEBS B Membranes

In a vial of HPLC-grade chloroform (20 mL), Br-Hex-SEBS **5** and functionalized PPO (DMA-Tri-PPO **1** or DMA-PPO **2**) were combined and agitated until a consistent solution was obtained. To the Br-Hex-SEBS **5**, the functionalized PPO was added at a 50 mol% ratio. After being heated at 40 °C overnight to crosslink the polymer solution, it was cooled to ambient temperature before the contaminants were filtered out. A glass Petri plate containing the solution was used to allow it to dry for 24 h at room temperature. After soaking the dried membranes in DI water and rinsing them to get rid of any leftover solvent, the membranes were able to be peeled off the dish. Trimethyl amine (TMA) solution was then applied to the membranes at 40 °C for 24 h, and the TMA was subsequently removed by washing the membranes with DI water. The membranes were then repeatedly rinsed with DI water before performing any tests after being submerged in 1 M KOH solution for at least 24 h at room temperature to facilitate the exchange of Br- and OH^−^ ions. The thickness of each polymer membrane was controlled by controlling the concentration of the polymer in the solvent used for the membrane fabrication.

### 2.9. Fabrication of Membrane Electrode Assemblies (MEAs) and Single-Cell Measurements

An MEA was prepared by decal transfer of catalyst layers (CLs) of 5 cm^2^ onto a mem-brane at 50 °C, 2 MPa (or metric ton) for 20 min with anion exchange membranes using a Heating Plate (QMESYS, QM900S) machine. For the anode CL, IrO_2_ Ink (Boyazenergy, Seoul, Republic of Korea) was prepared with 20 wt% FAA-3 ionomer/IrO_2_ and 90 wt% MeOH/IrO_2_ ratio. The resultant catalyst mixtures were dispersed by swing planetary mixture (Hantech, Gyeonggi-do, Republic of Korea) at 1800 rpm for 10 min, followed by sonication for 10 min., and then black ink was spray-coated onto the surface of the prepared membrane by using Ultrasonic Spray System (Boyazenergy, Seoul, Republic of Korea) followed by drying and annealing for CLs at 80 °C for 3 h. the IrO_2_ loading level was carefully controlled for the anode as 4.0 ± 0.02 mg/cm^2^. The cathode used was a Gas Diffusion Electrode (GDE) purchased from CNL energy, Seoul, Korea. Pt, which was spread out on Vulcan XC-72 carbon with a 40 wt% ratio, was dispersed on PTFC-treated GDL (Toray Carbon paper 120, 370 μm). Its Pt loading was 0.5 mg/cm^2^.

An MEA, 5 cm^2^ active area, was placed between a pair of gaskets, a Ni-coated monopolar plate with a single serpentine flow field, and two different porous transport layers (PTLs) (Ti-PTL for the anode, carbon fiber paper for the cathode). A 1 M KOH solution flowed on both the anode and cathode sides with a 10 mL/min flow rate. After 20 min for stabilization, polarization curves were recorded by scanning voltage from 1.4 to 2.0 V with 0.025 V/min for stabilization at 70 °C and ambient pressure.

## 3. Results and Discussion

### 3.1. Characterization and Production of DMA-Tri-PPO 1 and DMA-PPO 2

The following steps were used to create X-TriPPO-50SEBS **A** and x-PPO-50SEBS **B**, two AEMs created by crosslinking PPO and SEBS with triazole. First, dimethyl amine-triazole-functionalized PPO (DMA-Tri-PPO) **1** and dimethylamine-functionalized PPO (DMA-PPO) **2** were created. The following technique was followed to create DMA-Tri-PPO **1**. To create bromobenzylated PPO, also known as Br-PPO, the benzyl site of PPO was first grafted with bromine using NBS and AIBN (Figure 1). Therefore, PPO was grafted with 33 mol% Br. After that, polymer **3** and sodium azide were combined to create azidobenzylated PPO, or N_3_-PPO, **4**, which was then combined with 3-dimethylamino-1-propyne and PMDTA with the aid of CuBr(I) as a catalyst. DMA-Tri-PPO **1** was made as a result (Appendix A).

The following is how DMA-PPO **2** was made: to replace the Br in polymer **3** with dimethylamine, the previously described Br-PPO **3** was reacted with dimethylamine in an NMP solvent. Therefore, DMA-PPO **2** was produced (Appendix A).

Comparative spectroscopy using ^1^H NMR was used to determine the structure of the polymers produced in each stage (Appendix A). At 6.47 ppm, an aromatic proton peak was seen in PPO, while at 2.11 ppm, a benzyl proton peak (H_0_) could be seen. The aromatic proton signal shifted downfield, or the aromatic proton peak (H_4_) was observed at 6.70 ppm after the benzyl site was brominated. 

Moreover, a bromobenzylic proton peak (H_2_) showed up at 4.35 ppm, indicating that Br-PPO **3** was successfully synthesized. Based on the comparison of the relative intensities of H_3_, the aromatic proton peak of the original PPO, and H_4_, the aromatic proton peak created by the Br group, it was determined how many Br groups were added. As a result, it was determined that the Br groups inserted were roughly 33 moles per benzylic proton (Appendix A).

The bromobenzylic proton peak of Br-PPO **3** was shifted from 4.35 ppm to 4.22 ppm by the addition of the azide group. This discovery verified that N_3_-PPO **4** was successfully synthesized (Appendix A). It was also established that the click reaction caused the bromobenzylic proton peak to move from 4.22 ppm to 5.38 ppm. In addition, the methyl proton signal assigned to the amine group occurred at 2.31 ppm, and the proton peak corresponding to the carbon between the triazole and the amine group was discovered at 3.64 ppm. Triazole’s double-bond proton peak was noticed at a concentration of 7.54 ppm. It was determined that these peaks had a relative integral ratio of 2:6:1, demonstrating that DMA-Tri-PPO **1** was successfully synthesized (Appendix A).

The addition of dimethylamine caused the bromobenzylic proton peak of Br-PPO **3** to shift from 4.35 ppm to 3.39 ppm, providing evidence that DMA-PPO **2** was successfully synthesized (Appendix A).

### 3.2. Synthesis and Characterization of Br-Hex-SEBS 5

A functionalized SEBS to be crosslinked with DMA-Tri-PPO **1** and DMA-PPO **2** was created, bromohexyl SEBS (Br-Hex-SEBS) **5**. Initially, bromohexanoyl SEBS (Br-Hex-CO-SEBS) **6** was created by Friedel–Crafts acylation of SEBS with 6-bromohexanoyl chloride using AlCl_3_ as a catalyst. Here, the bromohexanoyl group was added to the styrene of SEBS at a mole ratio of roughly 50 mol%. Triethylsilane and TFA then reduced the carbonyl group of Br-Hex-CO-SEBS **6** to create Br-Hex-SEBS **5** (Appendix A).

Comparative spectroscopy employing ^1^H NMR was used to identify the structure of Br-Hex-SEBS (**5**) (Appendix A). After Friedel–Crafts acylation, the aromatic proton peaks (H_17_, H_18_) in compound **6** occurred, ranging from 7.40 ppm to 7.93 ppm. The carbonyl group had moved these peaks downfield. Further, the addition of bromine produced a proton peak (H_13_) at 3.43 ppm and a proton peak (H_12_) at 2.97 ppm that was assigned to the aliphatic carbon close to the carbonyl group. This verified that Br-Hex-CO-SEBS **6** had been successfully synthesized. 

Based on the relative integral ratio of the peaks assigned to the aromatic proton of styrene (H_14_, H_15_, H_16_, H_17_, and H_18_) and the H_13_ peak, the ratio of the bromohexanoyl groups added to the styrene of SEBS was determined. As a result, it was discovered that the added bromohexanoyl groups had a mole ratio of roughly 50% to styrene (Appendix A).

The signal attributable to the C-H bond (H_12_ at 2.97 ppm) next to the carbonyl group vanished after the reduction of the acyl group, and in compound **5**, a benzylic C-H peak (H_12_ at 2.56 ppm) formed. The aromatic proton peaks (H_17_, H_18_) of chemical **6** migrated upfield from 6.17 ppm to 7.21 ppm following the reduction procedure (Appendix A).

### 3.3. Fabrication of Crosslinked x-TriPPO-50SEBS A and x-PPO-50SEBS B Membranes

The following process was used to create crosslinked membranes made of chemically crosslinked PPO and SEBS: After being dissolved in CHCl_3_, DMA-Tri-PPO **1** and Br-Hex-SEBS **5** were cast onto membranes to cause cross-linking. The remaining bromine groups that had not been crosslinked were subsequently converted to quaternary ammonium (QA), while the counter-ions of QA were changed to OH^−^ ions by reacting with the resultant membrane with TMA. In the end, tri-azole-functionalized crosslinked PPO–SEBS membrane x-TriPPO-50SEBS A was created (Figure 1a).

Using a similar process, crosslinked membrane **B** was created by reacting DMA-PPO **2** and Br-Hex-SEBS **5** (Figure 1b).

The consistent thickness of these membranes (A and B) was discovered to be between 40 and 50 μm (Figure 2).

The constructed crosslinked membranes had QA as their ion-conducting head group. The IEC and ionic conductivity of the AEM should increase as there is more QA present (or as more QA is injected). However, high IEC values always cause the WU and SR of the membrane to increase. The “dilution effect” will cause a membrane’s ionic conductivity to decrease as its WU exceeds a particular threshold [52]. While creating a membrane electrode assembly, it is difficult for catalysts to be properly adsorbed onto the polymer membrane surface if the SR is too high (MEA).

Moreover, when an MEA such as this is used in a WE cell, the catalyst layers may delaminate when the membrane is put through a continuous cycle of expansion and contraction [53,54]. The IEC of the membrane will be reduced and ionic conductivity will be decreased if the amount of QA inserted into the polymer membrane is too small. Over time, this will cause decreased WE efficiency. To give the two crosslinked membranes in this work a suitable IEC, the mole ratio of the amine group of DMA-Tri-PPO **1** or DMA-PPO **2** to the bromine group of Br-Hex-SEBS **5** was fixed at 50 mol%.

To ascertain whether the crosslinking method was successful, the gel fraction of the two different crosslinked AEMs was assessed as an indirect indication of crosslinking. Both x-TriPPO-50SEBS (**A**) and x-PPO-50SEBS (**B**) had high gel fractions (100 and 99.1%, respectively), suggesting that both membranes were well crosslinked (Appendix A).

### 3.4. IEC, Ion Conductivity, WU, SR, and Density

The two crosslinked PPO–SEBS membranes’ fundamental physical characteristics were evaluated. The milliequivalent of the ion-conducting head groups present in a unit mass of an ion-conducting polymer membrane is referred to as the membrane’s IEC. The ionic conductivity of membranes made of ion-conducting polymer is also tightly correlated with this index. The ionic conductivity of a polymer membrane generally tends to increase with increasing IEC. Introducing several conducting groups into the polymer membrane is indicated by a high IEC. Moreover, the amount of water that can be contained by an ion-conducting polymer membrane increases with its IEC. In AEMs, water is used as a carrier to transport OH^−^ ions, directly improving ionic conductivity. Moreover, more conducting head groups enable the AEM to take up more water, increasing WU and SR. However, if the membrane absorbs too much water, it expands excessively and becomes mechanically unstable [53,54].

The number of moles of bromine groups present in Br-Hex-SEBS **5** and the total mass of the polymers used in membrane casting were used to compute theoretical IEC values. The values were established using the ^1^H NMR data’s H_14_ peak, which was assigned to bromine, and the relative integral ratio of the aromatic peaks (H_15_, H_16_, H_17_, H_18_, and H_19_) (Appendix A). x-TriPPO-50SEBS **A** and x-PPO-50SEBS **B** had theoretical IECs of 1.30 and 1.37 meq/g, respectively (Table 1).

With the use of back titration, the experimental IEC values of x-TriPPO-50SEBS **A** and x-PPO-50SEBS **B** were determined to be 1.17 and 1.25 meq/g, respectively (Table 1). The experimental IECs in both AEMs were marginally lower than the theoretical values, although the discrepancy was not very large. 

The two crosslinked membranes’ ionic conductivity was then assessed at 20 and 80 °C. Although x-TriPPO-50SEBS **A** displayed a lower IEC, the triazole-containing compound’s ionic conductivity was higher than that of x-PPO-50SEBS **B**. By dividing the ionic conductivity by the corresponding experimental IEC value, the normalized conductivity values were further estimated. In terms of normalized conductivity, x-TriPPO-50SEBS **A** and x-PPO-50SEBS **B** had a greater discrepancy. These findings proved that the x-TriPPO-50SEBS **A** membrane containing triazole had a higher water content relative to the amount of conducting head groups, which resulted in higher ionic conductivity. As a result, the triazole-containing x-TriPPO-50SEBS **A** membrane had a greater conduction efficiency than the x-PPO-50SEBS **B** membrane (Figure 3).

In addition, the normalized conductivity of these two crosslinked membranes was compared with that of other SEBS- and PPO-based AEMs (Appendix A). Both x-PPO-50 SEBS and x-TriPPO-50SEBS showed high normalized conductivity despite their relatively low IEC values, indicating the efficient ion transport of these crosslinked membranes.

In summary, a membrane’s WU and SR tend to increase with increasing IEC; however, x-TriPPO-50SEBS **A** displayed greater WU and SR values than x-PPO-50SEBS **B** while having a lower IEC. This was explained by the triazole group’s interaction with water, which led to the membrane retaining more water [55,56,57,58,59].

### 3.5. Mechanical and Thermal Properties

Besides acting as a polymer electrolyte to transport OH^−^ ions, an AEM also separates the oxygen produced at the anode during WE from the hydrogen (H_2_) produced at the cathode, increasing the quality of the hydrogen produced. Therefore, one of the most crucial elements affecting an AEM’s WE performance is mechanical stability. A comparison of the mechanical properties of x-TriPPO-50SEBS **A** and x-PPO-50SEBS **B** in terms of tensile strength and strain was done using the acquired stress–strain curves (Figure 4). x-TriPPO-50SEBS **A** had a tensile strength and strain of around 30 MPa and 110%, respectively. x-PPO-50SEBS **B** had a tensile strength and strain of roughly 20 MPa and 60%, respectively. Overall, x-TriPPO-50SEBS **A** with triazole outperformed x-PPO-50SEBS **B** in terms of strength and strain.

Another crucial criterion for WE at higher temperatures is the thermal stability of AEMs. TGA was used in this study to assess the thermal stability of x-TriPPO-50SEBS **A** and x-PPO-50SEBS **B**. (Appendix A). There were three stages of heat deterioration in both ion-conducting polymer membranes. The evaporation of the water present in the membrane was responsible for the weight loss seen between 30 and 200 °C. The breakdown of the quaternary ammonium group connected to the side chains of the polymer membrane was responsible for the weight loss that was noticed when the temperature approached 200 °C. The degradation of the polymer backbone was responsible for the final weight loss at temperatures higher than 350 °C [60,61]. The similarities between the weight loss patterns of x-TriPPO-50SEBS **A** and x-PPO-50SEBS B show that the two membranes have nearly equal levels of heat stability. Overall, the thermal stability of the two membranes was sufficient for WE.

### 3.6. Analysis of the Water Retention Capacity

Water has a large impact on how well OH^−^ ions conduct in AEMs. In general, there are two types of water that an AEM can absorb: bound water, which is bonded to ion-conducting groups, and free water, which is unaffected by ion-conducting groups. Ion-conducting groups in AEMs take advantage of positively charged QAs. The water absorbed by an AEM is separated into free water and bound water depending on how the QA and water interact. Free water is easily able to freeze or evaporate since it does not interact with the ion-conducting groups. By contrast, bound water has a higher boiling point and a lower freezing point than free water because of its interaction with ion-conducting groups.

The authors of this study had previously found that, rather than free water, OH^−^ ion conduction in AEMs was closely connected to bound water. Therefore, the quantity of water that contributes to ion conduction in an AEM is practically represented by its bound water content [62].

The ratio of free water to bound water in the two cross-linked AEMs was examined in the current study. The condition of the water in the AEMs was first evaluated using the TGA curves. The evaporation of the free water, which exhibited no reaction with the ion-conducting groups, was believed to be the cause of the weight loss seen below 100 °C.

The weight loss was attributed to the evaporation of bound water, whose boiling point was raised by its contact with the ion-conducting groups, in the temperature range between 100 and 200 °C, at the start of the degradation of ion-conducting groups.

This qualitative rather than quantitative TGA-based bound water analysis has some limitations (Figure 5a). The TGA results show that x-TriPPO-50SEBS **A** had a greater relative ratio of bound water than x-PPO-50SEBS **B**. Strong hydrogen bonds between the triazole’s nitrogen atoms and the water molecules were believed to be the cause of this behavior (Figure 5b).

To assess the membranes’ ability to retain water, DSC was used. In addition, bound-water analysis was carried out using DSC. Freezing and non-freezing bound water is separated into two categories in DSC [63]. Water that has had its freezing point decreased through interaction with ion-conducting groups is referred to as freezing-bound water. This kind of water freezes when it becomes colder than zero degrees. In non-freezing bound water, the water molecules are so close to the ion-conducting groups that they actively interact, preventing the water from freezing, even below 0 °C. Water has a greater impact on OH^−^ ion conduction the closer it is to ion-conducting groups. 

Hence, a higher ratio of non-freezing bound to freezing bound water represents a more conducive environment for ions. Endothermic melting peaks were seen in both AEMs in the DSC measurements (Figure 5c). There was only one endothermic melting peak in x-TriPPO-50SEBS **A** and two endothermic melting peaks in x-PPO-50SEBS **B**. All of the endothermic melting peaks seen in the two AEMs were caused by bound water that was frozen (dashed lines in Figure 5c). Because of interactions between water molecules and the nitrogen atoms of triazoles, which brought the water molecules closer to the ion-conducting groups, the melting peak of x-TriPPO-50SEBS **A** was lower than that of x-PPO-50SEBS **B**. With time, this resulted in a decreased melting point.

In x-PPO-50SEBS **B**, a further peak near 0 °C was seen. This peak is believed to be an endothermic melting peak caused by free water. Free-water peaks were not observed in x-TriPPO-50SEBS **A** because the volume of free water was insufficient to be recognized as a peak, even though it existed. Overall, x-TriPPO-50SEBS **A** had a higher relative ratio of bound water than x-PPO-50SEBS **B**. Because this free water and the freezing bound water could not be completely separated, freezing water is the term used to describe both forms of water. The difference between the freezing water content and the WU was then used to calculate the content of non-freezing water. For x-TriPPO-50SEBS **A** and x-PPO-50SEBS **B**, the proportion of non-freezing bound water was 80.64% and 47.81%, respectively (Figure 5d).

The WU of x-TriPPO-50SEBS **A** was higher than that of x-PPO-50SEBS **B** according to the TGA and DSC data. Moreover, in x-TriPPO-50SEBS **A**, both the relative ratio of frozen bound water to frozen water and the relative ratio of non-freezing bound water to frozen water were higher. Our findings demonstrated that triazole-containing x-TriPPO-50SEBS **A** retained more water than x-PPO-50SEBS **B**, resulting in enhanced ionic conductivity and increased WE efficiency.

### 3.7. Morphological Analysis

SEM, AFM, and contact angle analyses were used to examine the morphology of the two AEMs. According to the SEM pictures of the surfaces, the two crosslinked membranes had a homogeneous shape and no flaws (Figure 6a,d). Phase separation between the hydrophilic region—a darker area that corresponds to the ion-transporting channels—and the hydrophobic region—a lighter area—occurred in both crosslinked membranes according to the AFM-based surface study [64]. However, in x-TriPPO-50SEBS **A** as opposed to x-PPO-50SEBS **B**, the hydrophilic components of the surface were more intimately coupled (Figure 6b,e).

Conditions for OH^−^ ion conduction are more favorable when ion-transporting channels are more interconnected.

Contact angle study further demonstrated that the surface of x-TriPPO-50SEBS **A** was more hydrophilic than that of x-PPO-50SEBS **B**. The contact angles between x-TriPPO-50SEBS **A** and x-PPO-50SEBS **B** were 51.83° and 62.13°, respectively, showing that x-TriPPO-50SEBS **A** had a higher degree of hydrophilicity than x-PPO-50SEBS **B**. (Figure 6c,f).

The size of the ionic clusters was quantified and compared based on the d-spacings in SAXS analysis to assess microphase separation in both crosslinked membranes [65]. x-TriPPO-50SEBS **A** had d-spacings of 26.74, 7.12, and 4.50 nm, while x-PPO-50SEBS **B** had d-spacings of 16.34 and 4.14 nm (Figure 7).

Overall, x-TriPPO-50SEBS **A** had more peaks than x-PPO-50SEBS **B**, and the d-spacings were larger. This suggested that x-TriPPO-50SEBS A had larger, more organized ion-transporting channels. This was because of the triazole’s and water’s hydrogen bonds, which promoted the development of ionic channels. In addition, this outcome was in line with the AFM data phase separation characteristic.

### 3.8. Alkaline Stability

Another crucial element that affects how well AEMs work under alkaline conditions, or in WE’s actual operational circumstances, is their chemical stability. As already stated, the OH^−^ ions that AEM transports have a propensity to attack the primary backbone and the conducting groups, impairing the mem-characteristics of brane and ionic conductivity. This eventually causes WE’s cell performance to decline. The alkaline stability of x-TriPPO-50SEBS **A** and x-PPO-50SEBS **B** was investigated in this study. Over 960 h, changes in the ionic conductivity and IEC of the two AEMs were monitored while the membrane was submerged in a 1 M KOH solution at 80 °C.

A decrease in ionic conductivity during the entire measurement period for x-TriPPO-50SEBS **A** could not be observed. The conductivity lowering rate in x-PPO-50SEBS **B** was around 20%. (Figure 8a). This was attributed to the triazole’s nitrogen atoms, which allowed the ion-conducting polymer membrane to hold more water and efficiently defended the membrane from OH^−^ ion attacks. The literature has extensive information about how water acts as a barrier against OH^−^ ions.

Based on IEC measurements, the alkaline stability of the two crosslinked membranes was further investigated. The IEC of x-TriPPO-50SEBS **A** remained nearly unchanged in alkaline stability tests after 408 h and 960 h. However, the x-PPO-50SEBS B’s IEC declined at the same rate as that seen in the ionic conductivity measurements at 408 and 960 h (Figure 8b). Our findings demonstrated that the triazole-conjugated crosslinked membrane was chemically stable enough to support sustained WE operation.

### 3.9. Water Electrolysis (WE) Performance

Lastly, under the actual working conditions of AEMWEs, the single-cell performance of the two crosslinked AEMs was assessed. A 1 M KOH solution was allowed to flow through the cathode and anode at 70 °C while the thickness of x-TriPPO-50SEBS **A** and x-PPO-50SEBS **B** was fixed at 50 µm to measure cell performance. The current densities of x-TriPPO-50SEBS **A** and x-PPO-50SEBS **B** were 0.71 A/cm^2^ and 0.42 A/cm^2^, respectively, at a voltage of 1.8 V. (Figure 9). The triazole-containing x-TriPPO-50SEBS (**A**) membrane outperformed x-PPO-50SEBS (**B**). This was attributed to the effect of the triazole on the improved shape and ionic conductivity of the membrane.

In addition, the AEMWE durability of the x-TriPPO-50SEBS membrane, which showed better performance than the triazole-free crosslinked membranes developed in this study, was evaluated at 200 mA cm^−2^ and 70 °C. It was found that about 31 mV/h of voltage increase was observed for 280 min (Appendix A).

The WE cell performances of both PPO–SEBS-based AEMs were compared with those of other reported AEMs (Appendix A). Although direct comparison is not possible as cell performance depends on several factors, such as the catalysts, their loading levels, and back pressure, the x-TriPPO-50SEBS (**A**) membrane showed very good cell performance and hence is a good candidate for the WE application.

## 4. Conclusions

By chemically crosslinking two commercial polymers, PPO and SEBS, to create x-TriPPO-50SEBS, a crosslinked membrane containing triazole, and x-PPO-50SEBS, a crosslinked membrane devoid of triazole, this study created unique AEMs. Their ionic conductivity, alkaline stability, mechanical and physical characteristics, and single-cell performance during WE were tested and compared, particularly in light of the presence of triazole.

The triazole-containing polymer membrane displayed better shape and mechanical properties. During the whole temperature range, it had a greater ionic conductivity than the membrane lacking triazole. Moreover, x-TriPPO-50SEBS maintained nearly the same ionic conductivity and IEC even after 1000 h, with a retention rate of 99.6%, suggesting better chemical stability. When exposed to WE, it performed better in single cells (0.71 A/cm^2^ at 1.8 V) than its non-triazole counterpart.

Conclusively, despite having the same polymer structure, triazole gave one polymer membrane an improved shape compared to the other membrane by creating hydrogen bonds with water, which helped to improve membrane performance as a whole.

## Data Availability

Data is available upon request to the corresponding author.

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
