# Peer review of "Understanding the Effect of Triazole on Crosslinked PPO–SEBS-Based Anion Exchange Membranes for Water Electrolysis"

_polymers, 2023, doi:10.3390/polym15071736_

Round 1

Reviewer 1 Report

This work investigated connection of triazole-containing structure and membrane performance for WE. Membrane synthesis and characterization part is OK and compete in my point of view. However, some points need to be clearified.

1. The polarization curve of WE cell performance seems not that natural, I suggest to put more curves in that test (probably there will be more curves).

2. for WE cell test, why you choose 70oC, because most of work use 60 and some use 80.

3. 4mg cm-2 catalyst loading, whether it is too much for WE in 1M KOH and suggestion for furture work is that maybe you can use your own ionomer. FAA-3 seems not that fit to your membrane.

4. line 20 it is 0.7A/cm2, small mistake

Reviewer 2 Report

Title: Understanding the Effect of Triazole on Crosslinked PPO-SEBS-based Anion Exchange Membranes for Water Electrolysis
Ref. No.: 
polymers-2279377

Reviewer Comments:

This paper discussed the main component of electrolyzer which is AEM. This research field is important. But, this study required makes some revisions before its ready to publish. The specific comments are as follows,

1.     In abstract section, please add the problem statement of this study in this paper and the importance of this study.

2.     The introduction part should be written more professionally, with more up-to-date sources. The research gap should be communicated more clearly, along with the specific need for the completed study activity. What are the advantages of the AEMWE application? What is the importance of the triazole group? The advantages of this material compared to previous works also need to be discussed. Previous work related to this work should be discussed to highlight the significance of this study.

3.     Explain the mechanism of ionic conductivity of AEM.

4.     Explain the importance of thermal and mechanical properties of AEM for WE operating conditions.

5.     What is the requirement of AEM morphology in WE applications?

6.     Please makes comparison with previous study in term of AEM in table form.

Reviewer 3 Report

- The work is interesting and provides new data on the effect of triazole on crosslinked poly(phenylene  oxide) (PPO) and poly(styrene ethylene butylene styrene) (SEBS) membranes. The physicochemical including morphological and electrochemical properties of the membranes evaluated and their preliminary performance in anion exchange membrane water electrolysis (AEMWE) were evaluated. The article contains interesting data part worth publication. However, in my opinion, membrane preparation is such a complex process and most likely won’t be cheap. This goes against the idea of reducing the cost of such membranes to overcome the cost of fluorinated-based membranes.

Below are some remarks that need to be addressed before accepting for publication.

- A major part of the introduction is classical and contains major textbook materials as a background suitable for anion exchange membrane water electrolysis (AEMWE) as if the work is for the development of AEMWE system. More focus in the introduction should be given to the development of anion exchange membranes (AEM). Particularly, attention should be given to the significance of AEMs, their commercial status, various applications in AEMWE, desired properties in new membranes, and state of the art of research on the development of new membranes. It should be ended with a research gap prior to stating the research objective. The authors rather presented the work as an extension of previous work.

- The statement in lines 98-100 is confusing as one understand that both membranes are crosslinked while one of them containing triazine.

- The methods are scientifically sound and free of ambiguity. However, issues such as the method to control membrane thickness, how MEA was formed, and how the performance was tested should be presented in the main text (not in supplementary material).

- The accuracy of measured membrane properties such as thickness, water uptake, hydroxide-ion conductivity, chemical stability in KOH, etc., must be given.

- Line 124: DI water should be fully spelled as deionized (DI) water.

- What does it mean when the gel fractions of x-TriPPO- 32650SEBS A and x-PPO-50SEBSB were 100% and 99.1%? A comment statement should be added in this regard.

- It would be more convincing if the crosslinked triazole membrane properties are compared with one of the commercial AEMs such as Fumasep®, Sustainion®, AemionTM, or OrionTM. Data may be acquired from the literature following similarities in testing conditions. More importantly, the performance of the present membranes should be compared with the latest counterparts reported in literature to capture the main contribution of the present study.

- A summary of all membranes’ properties provides value to the study. This can be made by replacing Table 1.

- How long it took to produce polarization curves in Figure 9? In other words, how long would this performance remain?

- A statement addressing the membrane stability under dynamic condition should be given although accelerated stability test in KOH was given.

- The discussion on the properties and performance should be grounded by referring to your previous work on the same material and other studies from the literature.

- In the abstract and text: non-triazole counterpart should read the triazole-free counterpart.

Round 2

Reviewer 1 Report

Thank you for reply and now it can be accepted

Reviewer 2 Report

Authors have revised properly the reviewer comment. 

Reviewer 3 Report

The new version of the article captured the raised comments to a satisfactory level. The article should be accepted for publication.